# Increasing tendency of urine protein is a risk factor for rapid eGFR decline in patients with CKD: A machine learning-based prediction model by using a big database

Daijo Inaguma[1]*, Akimitsu Kitagawa[1], Ryosuke Yanagiya[2], Akira Koseki[3], Toshiya Iwamori[3], Michiharu Kudo[3‡], Yukio Yuzawa[4‡]

1 Department of Internal Medicine, Fujita Health University Bantane Hospital–Nagoya, Japan, 2 Division of Medical Information Systems, Fujita Health University School of Medicine–Toyoake, Japan, 3 IBM Research —Tokyo, Japan, 4 Department of Nephrology, Fujita Health University School of Medicine–Toyoake, Japan

☉ These authors contributed equally to this work.
‡ These authors also contributed equally to this work.
* daijo@fujita-hu.ac.jp

**Data Availability Statement:** 10.6084/m9.figshare. 12780311.

## Abstract

Artificial intelligence is increasingly being adopted in medical fields to predict various outcomes. In particular, chronic kidney disease (CKD) is problematic because it often progresses to end-stage kidney disease. However, the trajectories of kidney function depend on individual patients. In this study, we propose a machine learning-based model to predict the rapid decline in kidney function among CKD patients by using a big hospital database constructed from the information of 118,584 patients derived from the electronic medical records system. The database included the estimated glomerular filtration rate (eGFR) of each patient, recorded at least twice over a period of 90 days. The data of 19,894 patients (16.8%) were observed to satisfy the CKD criteria. We characterized the rapid decline of kidney function by a decline of 30% or more in the eGFR within a period of two years and classified the available patients into two groups—those exhibiting rapid eGFR decline and those exhibiting non-rapid eGFR decline. Following this, we constructed predictive models based on two machine learning algorithms. Longitudinal laboratory data including urine protein, blood pressure, and hemoglobin were used as covariates. We used longitudinal statistics with a baseline corresponding to 90-, 180-, and 360-day windows prior to the baseline point. The longitudinal statistics included the exponentially smoothed average (ESA), where the weight was defined to be $0.9^{*}(t/b)$, where t denotes the number of days prior to the baseline point and b denotes the decay parameter. In this study, b was taken to be 7 (7-day ESA). We used logistic regression (LR) and random forest (RF) algorithms based on Python code with scikit-learn library (https://scikit-learn.org/) for model creation. The areas under the curve for LR and RF were 0.71 and 0.73, respectively. The 7-day ESA of urine protein ranked within the first two places in terms of importance according to both models. Further, other features related to urine protein were likely to rank higher than the rest. The LR and RF models revealed that the degree of urine protein, especially if it exhibited an increasing tendency, served as a prominent risk factor associated with rapid eGFR decline.

**Funding:** IBM Research provided support for this study in the form of salaries for AK, TI and MK. The specific roles of these authors are articulated in the 'author contributions' section.

**Competing interests:** DI received lecture fees from Ono Pharmaceutical Co., Ltd. and Kyowa Hakko Kirin Co. YY received research support grants from Otsuka Pharmaceutical Co., Ltd., Kyowa Hakko Kirin Co., Ltd., and Chugai Pharmaceutical Co., Ltd. IBM Research provided support for this study in the form of salaries for AK, TI and MK. There are no patents, products in development or marketed products associated with this research to declare. This does not alter our adherence to PLOS ONE policies on sharing data and materials.

# Introduction

Chronic kidney disease (CKD) is a commonly occurring lifestyle-related disease. It induces problematic symptoms in patients [1], which can sometimes progress to end-stage kidney disease (ESKD) or cause cardiovascular (CV) disease. Its diagnosis is often delayed as most patients remain asymptomatic with respect to kidney dysfunction during stages 1, 2, and 3a of CKD. Therefore, medical check-ups and laboratory tests are essential not only for patients with diabetes or hypertension, but also for the general population. Several reports have demonstrated that treatment by a nephrologist could arrest the decline of estimated glomerular filtration rate (eGFR) in patients with CKD [2–4]. However, the ratio of nephrologists to patients with CKD is low across the world. Because of the reasons mentioned above, it might be helpful to identify patients with rapid decline of eGFR among the many CKD patients.

Previous large-scale cohort studies have identified several conditions, including proteinuria, hypertension, and comorbidity of diabetes, as risk factors associated with the rapid decline of eGFR [5–8]. Further, several clinical trials have established that reno-protective drugs such as renin angiotensin system blockers and sodium glucose transporter-1 inhibitors can decelerate the rate of eGFR decline, by comparing their effects with those of a placebo in CKD patients [9–12]. In other words, if we identify CKD patients exhibiting rapid decline in eGFR, we might be able to intervene the its course in an early stage. Recent studies have focused on evaluating kidney function trajectories in patients to predict the incidence of CV disease and all-cause mortality [13, 14]. Regardless of the primary kidney disease, the decline of eGFR is a common feature in patients with CKD. However, kidney function trajectories are often heavily patient-dependent. Previous reports have established that rapid eGFR decline is related to blood pressure-related problems, comorbidity, and proteinuria, not only in patients with CKD, but also in the general populace [15, 16].

Rapid development has been made in the field of artificial intelligence (AI) since the 1980s. In recent times, machine learning-based methods have found applications in various fields, including medicine [17–20]. In particular, artificial neural networks have been applied in nephrology for various prediction purposes [21–24]. The ability to automatically identify irregularities in data makes machine learning especially useful for big data comprising a large number of variables, where manual alternatives are not viable. Therefore, machine learning can be potentially applied to big medical data and the prediction of associated phenomena. Our hospital has maintained a big database of more than 900,000 patients treated for different diseases since 2004. To the best of our knowledge, no AI-based methods have been proposed yet to identify the aforementioned risk factors associated to rapid eGFR decline. In this study, we assumed that kidney function trajectories of patients would be informative and aid the diagnosis and subsequent treatment of CKD. Therefore, we developed a machine learning-based model to predict rapid eGFR decline in CKD patients by using a big hospital database.

# Materials and methods

## Dataset and population criteria

We constructed a database based on the information of 118,584 patients recorded by the electronic medical records system of the Fujita Health University Hospital during the period of June 2004 to July 2019. The database included the measured eGFRs for each patient, recorded at least twice over a period of 90 days. This study only used the data of 914,280 patients. 19,894 patients (16.8%) among them were observed to satisfy the following CKD criteria. In this study, CKD was defined to be characterized by eGFR $< 60$ ml/min/1.73 m$^2$ and/or urine protein $> 1+$, as determined by the dipstick method, over a period of more than 90 days.

Further, each patient was required to be at least 20 years old for measurement of eGFR and urine protein, and each previous measurement was required to have been recorded within two years of the current one. We excluded patients who had undergone dialysis or kidney transplantation before reference points. Information about comorbidity of diabetes, history of acute kidney injury (AKI), and use of renin angiotensin system inhibitor (RASI) was obtained from ICD-10 of electric medical records.

## Classification of patients based on trajectory of eGFR

The rapid decline (RD) of eGFR in CKD patients was defined to be a decline of 30% or more in eGFR within a period of 2 years [25–27]. As an accurate metric for the eGFR value, we used average eGFR measurements over a period of 90 days for each patient to avoid temporal spikes in data. Following this procedure, we identified 5,609 unique CKD patients exhibiting rapid eGFR decline and collected an aggregate of 9,866 samples from them. To form our cohort, in addition to the 9,866 RD samples from 5,609 unique patients, we created control (non-RD) samples by extracting eGFR trajectories from patients with similar profiles exhibiting (1) non-RD eGFR, (2) rapid eGFR decline beginning less than 2 years before the positive sample, (3) same gender, or (4) least mean average difference between ages and eGFR values at the beginning of the trajectories. Following this procedure, we identified 4,302 unique control patients with CKD and extracted 9,866 samples not exhibiting rapid eGFR decline. Fig 1 shows the patient flow. Finally, we combined the two groups and identified an aggregate of 9,911 unique patients for the present study. Fig 2 indicates representative examples of reference points in each group. In some cases, the reference points have been set several times. The reference points of prediction for patients in either group, RD detection points for patients in the RD group, and measurement points where eGFR was used for matching for patients in the non-RD group were available.

## Predictive model

By assigning positive labels to samples in the RD group and negative labels to samples in the non-RD group, we constructed predictive models based on two machine learning algorithms. Longitudinal laboratory test results, including urine protein, blood urea nitrogen, systolic blood pressure, diastolic blood pressure, total cholesterol, hemoglobin, uric acid, and triglyceride, were taken to be the covariates. First, we noted the reference point values of the aforementioned tests, i.e., the latest values observed corresponding to the reference point of prediction. Only the baseline values of blood urea nitrogen were included. Next, we recorded the longitudinal statistics based on the past 90-, 180-, and 360-day windows from the reference point. The longitudinal statistics considered in this study were mean, standard deviation, and exponentially smoothed average (ESA), where the weight was defined to be 0.9*(t/b), where t is the number of days from the reference point and b is the decay parameter. In this study, b was taken to be 7 (weekly decay). The missing values corresponding to each laboratory test were imputed via the last observation carried forward method. If no data were available for a test, the mean value of the corresponding training data was used instead. Additionally, all the values were standardized. In the following step, by using the aforementioned covariates and training labels, we applied the logistic regression (LR) and random forest (RF) algorithms based on Python code and the scikit-learn library (https://scikit-learn.org/) to create two classification models for RD. We optimized the models by fine-tuning the hyperparameters of the algorithms, including the regularization parameters of LR and the number of trees, etc. for RF. After identifying the optimal parameters via inner four-fold cross validation, we evaluated these models using outer five-fold cross validation. The contribution to RD was evaluated via

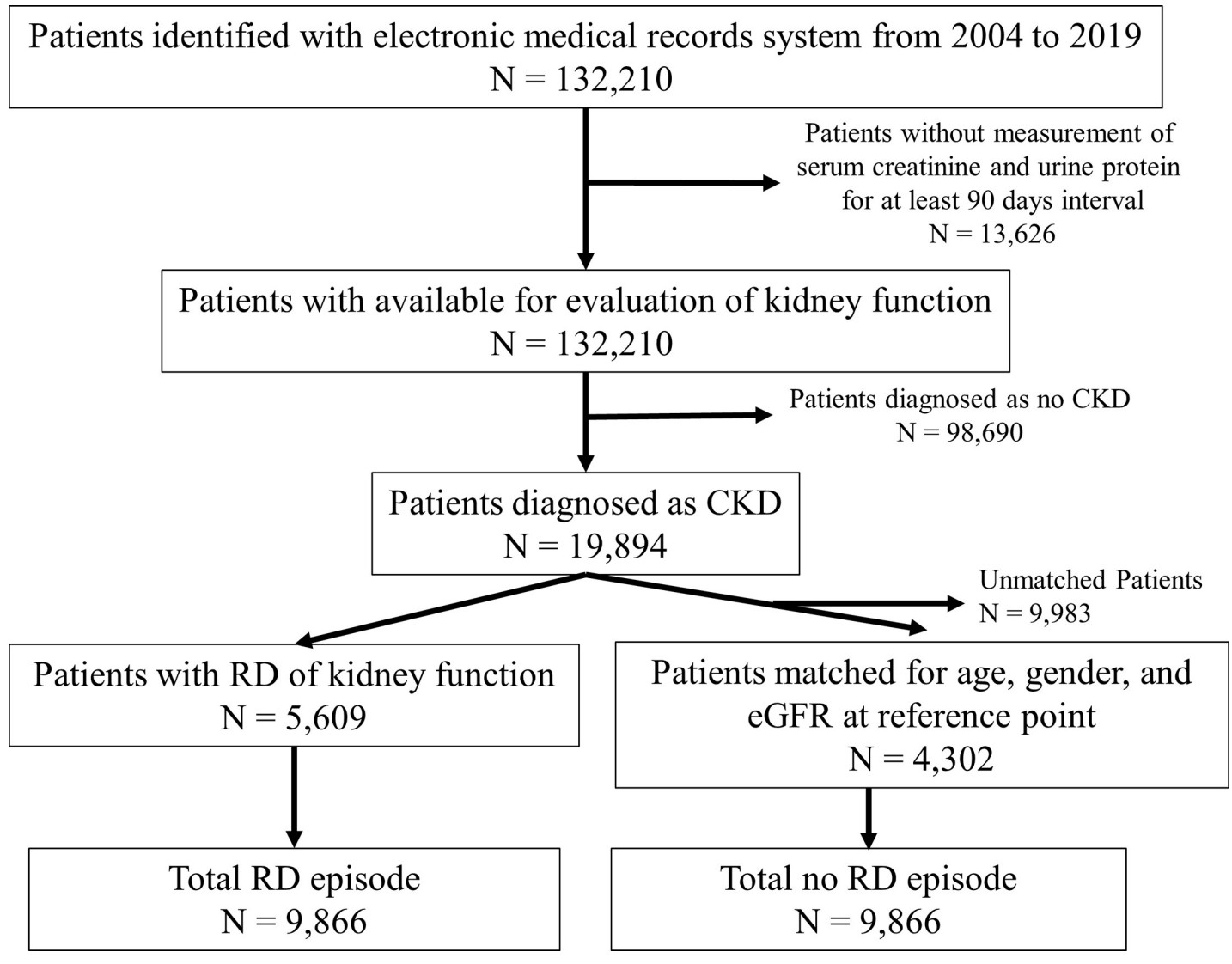

**Fig 1. Patient flow.**

the analysis of coefficient weights in the LR model and the decrease in mean Gini impurity in the RF model. We defined three patterns by grouping together the features as follows. Pattern 1 comprised comorbidity of diabetes, history of AKI, systolic blood pressure, diastolic blood pressure, use of RASIs, urine protein, hemoglobin, serum uric acid, blood urea nitrogen, serum total cholesterol, and serum triglyceride at the reference point; Pattern 2 comprised comorbidity of diabetes, history of AKI, systolic blood pressure, diastolic blood pressure, use of RASIs, urine protein, hemoglobin, serum uric acid, blood urea nitrogen, serum total cholesterol, and serum triglyceride at the reference point, for a period of 180 days, and the 7-day ESA of features prior to the reference point; and Pattern 3 comprised comorbidity of diabetes, history of AKI, systolic blood pressure, diastolic blood pressure, use of RASIs, urine protein, hemoglobin, serum uric acid, blood urea nitrogen, serum total cholesterol, and serum triglyceride at the reference point, for periods of 90, 180, and 360 days, and 7-day ESA of features prior to the reference point.

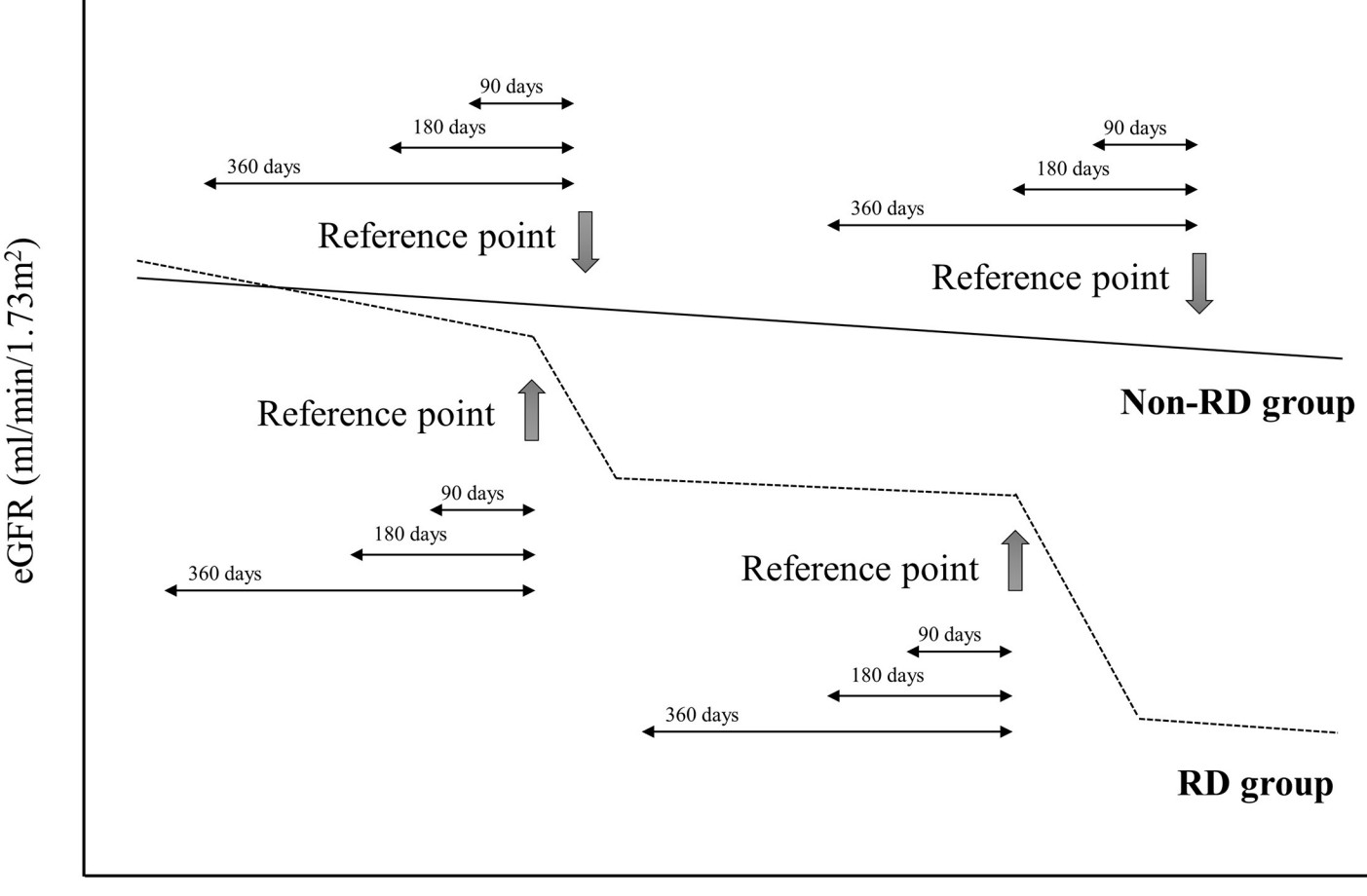

**Fig 2. Representative examples of reference points in each group.**

### Ethics approval and consent to participate

The present study was conducted following the Ethical guidelines for Clinical Research by the Japanese Ministry of Health, Labor, and Welfare (created July 30, 2003; full revision December 28, 2004; full revision July 31, 2008) and the Helsinki Declaration (revised 2013). It was approved by the clinical research ethics committees at Fujita Health University School of Medicine (approval number: HM19-157). All data were fully anonymized before we analyzed. The contents of the entire research have been displayed in the information disclosure document on the Web and Informed consent was obtained in the form of opt-out on the web-site. Those who rejected were excluded. The trial registration number of the study is UMIN 000037476, and it was registered on August 1, 2019.

## Results

### Comparison of patient characteristics and laboratory data at the reference point

Table 1 compares the patient characteristics and laboratory data of the two groups at the reference point. No significant differences in age, sex, eGFR, and rate of history of acute kidney

**Table 1. Patients characteristics and laboratory data at reference point.**

| Variables | All n, 19,732 | RD group n, 9,866 | Non-RD group n, 9,866 | p value |
|---|---|---|---|---|
| Age (years old) | 68.5, 13.7 | 68.5, 13.7 | 68.5, 13.6 | 1.000 |
| Female gender (%) | 41.7 | 41.7 | 41.7 | 1.000 |
| Comorbidity of diabetes (%) | 31.3 | 35.6 | 27.1 | < 0.001 |
| History of AKI (%) | 4.6 | 4.6 | 4.5 | 0.707 |
| SBP (mmHg) | 131, 26 | 136, 26 | 128, 26 | < 0.001 |
| DBP (mmHg) | 73, 15 | 74, 15 | 72, 15 | < 0.001 |
| Use of RASIs (%) | 61.8 | 56.8 | 66.8 | < 0.001 |
| eGFR (ml/min/1.73m$^2$) | 39.9, 26.0 | 39.9, 26.0 | 39.9, 26.1 | 0.760 |
| Serum creatinine (mg/dL) | 2.23, 2.04 | 2.25, 2.05 | 2.21, 2.04 | 0.061 |
| BUN (mg/dL) | 29.5, 19.1 | 29.8, 17.9 | 29.3, 20.1 | < 0.001 |
| Hemoglobin (mg/dL) | 11.5, 2.2 | 11.4, 2.1 | 11.5, 2.3 | 0.001 |
| Hematocrit (%) | 34.8, 6.4 | 34.7, 6.1 | 34.9, 6.8 | < 0.001 |
| Serum T-C (mg/dL) | 181, 49 | 186, 50 | 175, 47 | < 0.001 |
| Serum TG (mg/dL) | 142, 91 | 151, 100 | 133, 79 | < 0.001 |
| Serum uric acid (mg/dL) | 6.2, 2.0 | 6.3, 1.9 | 6.0, 2.0 | < 0.001 |
| Urine protein * | 1.9, 1.8 | 2.3, 1.9 | 1.4, 1.6 | < 0.001 |
| Urine protein ** | 2 [0, 3] | 2 [0, 5] | 1 [0, 3] | |

Mean, standard deviation, Value, %

* Continuous value of urine protein test by dipstick

** Semi-quantity test of urine protein test by dipstick 50% [25%, 75%]

0; -, 1; ±, 2; +, 3; ++, 4; +++, 5; ++++

RD; rapid decline, AKI; acute kidney injury, SBP; systolic blood pressure, DBP; diastolic blood pressure, RASI; renin angiotensin system inhibitor, eGFR; estimated glomerular filtration rate, BUN; blood urea nitrogen, T-C; total cholesterol, TG; triglyceride

injury (AKI) were observed between the two groups. However, comorbidity of diabetes, blood pressure, serum total cholesterol, serum uric acid, serum triglyceride, and amount of urine protein were observed to be higher in patients in the RD group. Meanwhile, use of renin angiotensin system inhibitors was low in the RD group. Further, blood pressure, serum total cholesterol, serum uric acid, serum triglyceride, and amount of urine protein were higher in patients in the RD group over periods of 90, 180, and 360 days prior to the reference point (S1 Table) and the 7-day ESA (S2 Table).

## Comparison of areas under the curve (AUCs) of the two models

Fig 3 shows receiver operating characteristic curve and Table 2 compares the AUCs exhibited by the LR-based and the RF-based model in the prediction of RD. The AUCs exhibited by the LR-based model using the Pattern 1, 2, and 3 were observed to be 0.67, 0.69, and 0.71, respectively. In contrast, the AUCs exhibited by the RF model using the Pattern 1, 2, and 3 were observed to be 0.68, 0.71, and 0.73, respectively. The AUCs exhibited by both models were observed to increase with the increase in the number of features.

## Ranking of features according to the LR-based and RF-based models

The Pattern1 comprised eight 11 features—urine protein, systolic blood pressure, serum uric acid, blood urea nitrogen, serum total cholesterol, use of RASIs, hemoglobin, serum triglyceride, comorbidity of diabetes, diastolic blood pressure, and history of AKI—in order of importance as measured by the LR-based model. In contrast, the RF-based model provided the

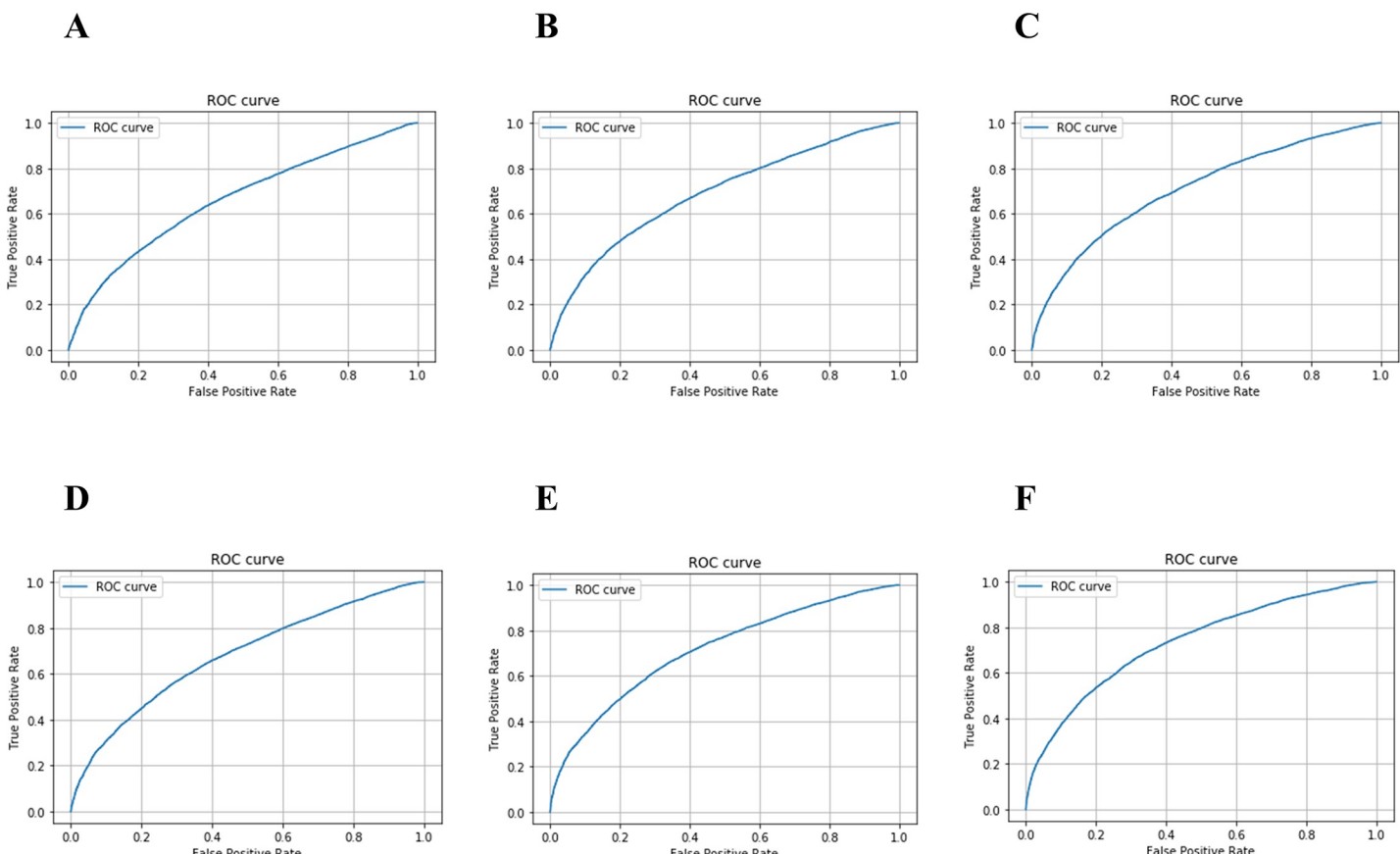

**Fig 3. Receiver operating characteristic curve for prediction of the RD. A.** The Pattern 1 (the LR model). **B.** The Pattern 2 (the LR model). **C.** The Pattern 3 (the LR model). **D.** The Pattern 1 (the RF model). **E.** The Pattern 2 (the RF model). **F.** The Pattern 3 (the RF model).

**Table 2. Comparison of AUC by models.**

| Model | Pattern | AUC |
|---|---|---|
| Logistic regression model | 1 | 0.67 |
| | 2 | 0.69 |
| | 3 | 0.71 |
| Random forest model | 1 | 0.68 |
| | 2 | 0.71 |
| | 3 | 0.73 |

Each feature includes: comorbidity of diabetes, history of AKI, SBP, DBP, use of RASIs, urine protein, hemoglobin, serum uric acid, BUN, serum total cholesterol, serum triglyceride

1; at baseline (at start point of rapid eGFR decline)

2; at baseline, average and standard deviation of features during 180 days prior to the baseline, and 7-day exponentially smoothed average of features

3; at baseline, average and standard deviation of features during 90, 180, and 360 days prior to the baseline, and 7-day exponentially smoothed average of features

AUC; area under curve, AKI; acute kidney injury, SBP; systolic blood pressure, DBP; diastolic blood pressure, RASI; renin angiotensin system inhibitor, BUN; blood urea nitrogen, eGFR; estimated glomerular filtration rate

**Table 3. Ranking of 10 top logistic regression and random forest model features.**

| Rank | Logistic regression | | Random forest | |
|---|---|---|---|---|
| Features | 2 | 3 | 2 | 3 |
| 1 | hemoglobin (7-day ESA) | urine protein (7-day ESA) | urine protein (7-day ESA) | hemoglobin (90 SD) |
| 2 | urine protein (7-day ESA) | hemoglobin (7-day ESA) | hemoglobin (180 SD) | urine protein (7-day ESA) |
| 3 | hemoglobin (180 mean) | SBP (7-day ESA) | urine protein (180 mean) | urine protein (180 mean) |
| 4 | total cholesterol (baseline) | hemoglobin (90 SD) | urine protein (baseline) | urine protein (360 mean) |
| 5 | hemoglobin (180 SD) | total cholesterol (baseline) | uric acid (180 SD) | urine protein (90 mean) |
| 6 | SBP (7-day ESA) | total cholesterol (7-day ESA) | uric acid (7-day ESA) | hemoglobin (180 SD) |
| 7 | total cholesterol (7-day ESA) | hemoglobin (360 mean) | uric acid (180 mean) | urine protein (baseline) |
| 8 | SBP (180 mean) | hemoglobin (180 mean) | total cholesterol (baseline) | hemoglobin (360 SD) |
| 9 | urine protein (180 mean) | hemoglobin (90 mean) | BUN (baseline) | total cholesterol (90 SD) |
| 10 | hemoglobin (baseline) | uric acid (90 SD) | SBP (baseline) | uric acid (90 SD) |

Features

2; at baseline, average and standard deviation of features during 180 days prior to the baseline, and 7-day ESA of features

3; at baseline, average and standard deviation of features during 90, 180, and 360 days prior to the baseline, and 7-day ESA of features

ESA; exponentially smoothed average, SBP; systolic blood pressure, SD; standard deviation, BUN; blood urea nitrogen

following list in order of importance: urine protein, systolic blood pressure, serum total cholesterol, blood urea nitrogen, serum uric acid, hemoglobin, serum triglyceride, diastolic blood pressure, use of RASIs, comorbidity of diabetes, and history of AKI comprised. Table 3 lists the top-10 ranking of features comprising the Pattern 2 and 3, according to both models. Significantly, the 7-day ESA of urine protein ranked within the first two places corresponding to both models. Further, features related to urine protein were observed to be more likely to rank higher than the rest. The 7-day ESA of hemoglobin was also consistently placed at high ranks corresponding to the LR-based model.

## Discussion

We demonstrated that proteinuria, especially when it exhibited a recent spike, was important in the prediction of rapid eGFR decline in CKD patients being treated in a hospital. The present study exhibited three primary characteristics. First, we analyzed big data via machine learning algorithms. We also adopted the ESA of variables as the primary metric during the extraction of risk factors because we considered the long-term trends of each variable, as they are meaningful in the prediction of eGFR trajectory. Second, we adopted ESA as one of the features, containing, in particular, the ESA of urine protein. This enabled us to weigh features closer to the reference point. Finally, the subjects in the present study included out-patients suffering from various diseases involving CKD, while a certain proportion of the data contained kidney function reports from different sections of population, including the general population, elderly population, or patients already diagnosed with CKD. The primary cause of CKD cannot be narrowed to a single kidney disease in many cases, as more often than not, the symptoms are caused by complications arising from a combination of two or more diseases. Diseases other than kidney-related ones can also sometimes lead to CKD directly or indirectly during the follow-up period. As the hospital considered in this study is the biggest in Japan, we had access to data pertaining to a large number of patients suffering from various diseases. Because of the diversity of available data, it can be concluded that the results of the present study are informative to manage patients who are needed to be followed for different diseases in large-sized hospitals. In predictive models to see an unknown future using past data, AUC

around 0.7 is generally regarded as being good, and an improvement from 0.71 (LR-based model) to 0.73 (RF-based model) was thought to be rather remarkable.

Besides proteinuria, the present study also established the ESA of proteinuria to be one of the most prominent risk factors associated to rapid eGFR decline. This corroborates the conclusions of several other cohort studies, which have indicated that proteinuria is significantly associated with certain kidney function metrics, including the doubling of serum creatinine level, eGFR halving, and progression to ESKD [5, 6]. The Clinical Renal Insufficiency Cohort (CRIC) study, conducted in the United States of America, calculated the hazard ratios for ESKD and eGFR halving corresponding to the highest and the lowest proteinuria categories to be 11.83 and 11.19, respectively [5]. Meanwhile, the Chronic Kidney Disease Japan Cohort (CKD-JAC) also reported that increased albumin-to-creatinine ratio at the baseline was significantly associated to eGFR halving and progression to ESKD at the primary end-point [6]. It was crucial to carefully observe the temporal trends of urinary protein excretion predict eGFR decline as early as possible. We consider the aforementioned conclusions of the study to be novel and informative. Meanwhile, it was reported that eGFR decline in patients with CKD stage 3 was relatively slow[28] and episode of AKI generally affects trajectories of eGFR [29, 30]. Hence, we used episode of AKI as a variable in the analysis. However, the episode of AKI did not rank in the top-10 in terms of feature importance. We considered that the reason might be due to low incident rate of AKI in the present study.

AI-based prediction has been attempted in various medical fields, especially in nephrology [22, 31–33]. Nationwide studies and cohorts have been conducted all over the world to analyze big data more effectively. We accumulated more than 132,000 pieces of medical data, ranging from 2004 to the present from a single hospital. The prediction accuracy of the proposed methods can be further improved by appending additional parameters, such as average and standard deviation values prior to the reference time, and by complementing textual data with digital data despite the retrospective study design. Moreover, the analysis of ESA, which was established to be the most prominent feature by the present study, was only possible due to the application of machine learning. We consider the conclusions of the study to be of use in real-world clinical scenarios despite the preliminary nature of the study.

The study had the following limitations. First, patient information, including medical histories, comorbidities, and medications through ICD-10 code, was not completely available. This was because some patients opted for concurrent treatment of other diseases at other hospitals. Second, the intervals between successive examinations or the frequency of examinations, including blood tests, were dependent on individual patients. Hence, we used the average values over periods of 90, 180, and 360 days prior to the baseline. Finally, even though urinary protein creatinine ratio is currently the best metric for the evaluation of kidney disease, the measurement of proteinuria was only available via the semi-quantitative method using dipsticks in the present study. This is because the measurement of urinary protein creatinine ratio has only become popular among physicians other than nephrologists over the last decade. Based on the aforementioned limitations, it is evident that the use of more advanced systems to acquire more detailed information, including medications prescribed at other facilities, is necessary to enhance the accuracy of the proposed rapid eGFR decline prediction methods.

## Conclusion

The proposed RF-based and LR-based models based on machine learning algorithms were proved to be effective in the identification of patients with rapid eGFR decline in real-world clinical scenarios. Further, urine protein, especially if it exhibited a recent spike, was established to be a prominent risk factor associated with rapid eGFR decline.

## Supporting information

**S1 Table. Average blood pressure and laboratory data for 90, 180, and 360 days prior to the baseline.**
(DOCX)

**S2 Table. Exponentially smoothed average of blood pressure and laboratory data for 7 and 30days.**
(DOCX)

## Author Contributions

**Conceptualization:** Daijo Inaguma.

**Data curation:** Akimitsu Kitagawa, Akira Koseki, Toshiya Iwamori.

**Formal analysis:** Akira Koseki, Toshiya Iwamori.

**Investigation:** Akira Koseki.

**Methodology:** Akira Koseki, Toshiya Iwamori.

**Project administration:** Ryosuke Yanagiya.

**Software:** Akira Koseki.

**Supervision:** Michiharu Kudo, Yukio Yuzawa.

**Writing – original draft:** Daijo Inaguma, Akimitsu Kitagawa, Akira Koseki.

**Writing – review & editing:** Daijo Inaguma.

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
