## [Decision Letter · Decision Letter 0]

1 Jul 2020

PONE-D-20-15213

A machine learning-based prediction model for rapid glomerular filtration rate decline in patients with chronic kidney disease by using a big database

PLOS ONE

Dear Dr. Inaguma,

Thank you for submitting your manuscript to PLOS ONE. After careful consideration, we feel that it has merit but does not fully meet PLOS ONE’s publication criteria as it currently stands. Therefore, we invite you to submit a revised version of the manuscript that addresses the points raised during the review process.

The authors should incorporate important confounding factors in analysis as suggested by the expert.

We look forward to receiving your revised manuscript.

Kind regards,

Tatsuo Shimosawa, M.D., Ph.D.

Academic Editor

PLOS ONE

Journal Requirements:

2. In the ethics statement in the manuscript and in the online submission form, please provide additional information about the patient records used in your retrospective study. Specifically, please ensure that you have discussed whether all data were fully anonymized before you accessed them and/or whether the IRB or ethics committee waived the requirement for informed consent. If patients provided informed written consent to have data from their medical records used in research, please include this information.

4.We note that you have indicated that data from this study are available upon request. PLOS only allows data to be available upon request if there are legal or ethical restrictions on sharing data publicly. For information on unacceptable data access restrictions, please see http://journals.plos.org/plosone/s/data-availability#loc-unacceptable-data-access-restrictions.

5.Thank you for stating the following in the Competing Interests section:

[DI received lecture fees from Ono Pharmaceutical Co., Ltd. and Kyowa Hakko Kirin Co. YY received research support grants from Otsuka Pharmaceutical Co., Ltd., Kyowa Hakko Kirin Co., Ltd., and Chugai Pharmaceutical Co., Ltd. This does not alter our adherence to PLOS ONE policies on sharing data and materials.].   

We note that one or more of the authors are employed by a commercial company: IBM Research

Reviewers' comments:

Reviewer's Responses to Questions

**Comments to the Author**

1. Is the manuscript technically sound, and do the data support the conclusions?

Reviewer #1: Partly

Reviewer #2: Partly

2. Has the statistical analysis been performed appropriately and rigorously? 

Reviewer #1: No

Reviewer #2: I Don't Know

3. Have the authors made all data underlying the findings in their manuscript fully available?

Reviewer #1: No

Reviewer #2: Yes

4. Is the manuscript presented in an intelligible fashion and written in standard English?

Reviewer #1: Yes

Reviewer #2: No

5. Review Comments to the Author

Reviewer #1: In this study, the authors investigated the risk factors for the rapid decline in estimated glomerular filtration rate (eGFR) using logistic (LR) and random forest (RF) models. They found that urinary protein level and its tendency to increase were associated with rapid eGFR decline. Although potentially useful and interesting, I have several statistical concerns.

1. Title: I think that the most important message to readers is that the presence of urine protein is a risk factor for the rapid decline in eGFR, as indicated by RF. The title “A machine learning-based prediction model for rapid…..” does not reflect the contents of this paper. How about changing the title considering above?

2. Endpoint, P. 8: The endpoint was defined as “a decline of 30% or more in eGFR within a period of 2 years”. How were the patients, who developed end-stage kidney disease or died within one year, treated? If they were not included in the analysis, there was a bias in the analysis.

3. Variables, P. 12: Causes of CKD such as diabetes mellitus, history of diseases such as cardiovascular disease, and use of medications such as angiotensin II receptor blockers use are necessary variables when investigating the risk factors for the rapid eGFR decline. Additional analysis including these variables should be conducted.

4. AUCs, P. 14: Table 2 shows the AUCs of the models. The AUCs of the RF models, about 0.7, were very low as the results of machine learning models. Because the values were almost the same as those of the LR models, there was no merit of machine learning analysis. It would be better to find new risk factors for the rapid decline in eGFR decline using usual statistical method properly.

Reviewer #2: This paper address is an important issue.It Reinforces findings of others. That proteinuria is the main determinant of the speed with which kidney function declines.It is not a random sample of patients they are hospitalized patients but it is unclear if the data is inpatient or outpatient or both that must be clarified.Inpatient data can be confounded by acute illnesses and their effect on kidney function

The CKD prevalence 16.8% which is considerably higher than average prevalence in the general population, 10%. The 28% that had greater than 30% decline in two years seems awfully high and again this is not a representative population since these patients were either hospitalized or come from patients in a hospital data base

.Regarding dipstick protein as the source of proteinurea,Interpretation of this needs to include urine specific gravity since the concentration of the urine can greatly affect the urine protein test

.Line 251 to 253 says the results apply to real world clinical settings I think that needs to be adjusted to say that these findings would apply to similar populations as the one that the study

. Line 268 to 2 74 is very confusing it needs to be rewritten.The strong correlation between the ESA of proteinuria and GFR decline established in the present

269 study was proved under the assumption that the increase in urinary protein excretion reflected the

270 exacerbation of glomerular hypertension and sclerosis. Other than proteinuria, the RF-based

271 model revealed that the ESA of serum creatinine level was also ranked in the top-10 in terms of

272 feature importance. Therefore, it was crucial to carefully observe the temporal trends of urinary

273 protein excretion and kidney function to predict GFR decline as early as possible. We consider

274 the aforementioned conclusions of the study to be novel and informative.

i do not understand this!!

Other factors that need to be discussed when evaluating the conclusions is the fact that episodes of AKI can affect the rate of change of kidney function overtime and the fact that stable renal function in a population of this age is very common. Reference Erikson KI 2006 ,Population of similar age where over a 10 year interval 27% had no decline kidney function.Another Issue that needs to be addressed in these types of studies is the competitive risk of death versus decline of kidney function and how that was adjusted for in this analysis

6. PLOS authors have the option to publish the peer review history of their article (what does this mean?). If published, this will include your full peer review and any attached files.

Reviewer #1: No

Reviewer #2: **Yes: **Steven Rosansky

---

## [Author Response · Author response to Decision Letter 0]

20 Aug 2020

Reviewer #1:

1. Title: I think that the most important message to readers is that the presence of urine protein is a risk factor for the rapid decline in eGFR, as indicated by RF. The title “A machine learning-based prediction model for rapid…..” does not reflect the contents of this paper. How about changing the title considering above?

Reply: Thank you for your suggestion. We changed the title of our article as it showed the contents.

2. Endpoint, P. 8: The endpoint was defined as “a decline of 30% or more in eGFR within a period of 2 years”. How were the patients, who developed end-stage kidney disease or died within one year, treated? If they were not included in the analysis, there was a bias in the analysis.

Reply: Thank you for valuable comments. We enrolled all patients who had not received maintenance dialysis before the reference points. In other words, the patients, who developed end-stage kidney disease or died within one year after the reference points were included in the present study. We revised the description (page 7, line 103 – 104)

3. Variables, P. 12: Causes of CKD such as diabetes mellitus, history of diseases such as cardiovascular disease, and use of medications such as angiotensin II receptor blockers use are necessary variables when investigating the risk factors for the rapid eGFR decline. Additional analysis including these variables should be conducted.

Reply: Thank you for reasonable comments. As you pointed out, variables including comorbidity of diabetes, use of renin angiotensin system inhibitors, and history of acute kidney injury were reported to be risk factors for rapid GFR declining. Therefore, we added the factors and re-analyzed. We show the results in the Table 1, 2, and add the description (page 7, line 104 – page 8, 106, page 10, line 152 – 159, page 20, line 290 – page 21, line 294). However, the variables did not rank 10 top features.

4. AUCs, P. 14: Table 2 shows the AUCs of the models. The AUCs of the RF models, about 0.7, were very low as the results of machine learning models. Because the values were almost the same as those of the LR models, there was no merit of machine learning analysis. It would be better to find new risk factors for the rapid decline in eGFR decline using usual statistical method properly.

Reply:

We conducted both logistic regression and random forest model by machine learning. Different from descriptive models which can marks very high, this paper discusses predictive models. In predictive models to see an unknown future using past data, AUC around 0.7 is generally regarded as being good, and an improvement from 0.71 (logistic regression) to 0.73 (random forest) is thought to be rather remarkable. Hence, we believe showing Random Forest results including prediction performance and its important factors compared to logistic regression is of interest to researchers of relevant disciplines. We added the description in the Discussion (page 19, line 269 - 272).

Reviewer #2: This paper address is an important issue.It Reinforces findings of others. That proteinuria is the main determinant of the speed with which kidney function declines.It is not a random sample of patients they are hospitalized patients but it is unclear if the data is inpatient or outpatient or both that must be clarified.Inpatient data can be confounded by acute illnesses and their effect on kidney function.

Reply:

Thank you for valuable comments. Proteinuria is one of the strongest risk factors for declining GFR in CKD patients. In addition, it often increases under sick conditions as you pointed out. In the present study, we used data from both outpatients and hospitalized patients. Therefore, we decided to use data such as eGFR, hemoglobin, and urine protein of mean values of 90, 180, and 360 days before the reference points in order to exclude transient changes of GFR caused by acute illness. In addition, we added and re-analyzed history of acute kidney injury as a feature (page 7, line 104 – page 8, 105, page 10, line 152 – 159, page 20, line 290 – page 21, line 294).

The CKD prevalence 16.8% which is considerably higher than average prevalence in the general population, 10%. The 28% that had greater than 30% decline in two years seems awfully high and again this is not a representative population since these patients were either hospitalized or come from patients in a hospital data base.

Reply:

As you pointed out, patients in our study were not representative because the database were from hospital-wide database. However, we thought that there were few reports from that database and therefore, it was one of valuable features, instead. We revised the description in the Discussion (page 19, line 268 - 269).

.Regarding dipstick protein as the source of proteinurea,Interpretation of this needs to include urine specific gravity since the concentration of the urine can greatly affect the urine protein test

Reply:

We totally agreed with you. Unfortunately, data of urine specific gravity was not available. Hence, we described that in the limitation (page 21, line 310 – page 22, line 313).

.Line 251 to 253 says the results apply to real world clinical settings I think that needs to be adjusted to say that these findings would apply to similar populations as the one that the study

Reply:

Thank you for your suggestion. We re-wrote the description (page 19, line 268 - 269).

. Line 268 to 274 is very confusing it needs to be rewritten.The strong correlation between the ESA of proteinuria and GFR decline established in the present 269 study was proved under the assumption that the increase in urinary protein excretion reflected the 270 exacerbation of glomerular hypertension and sclerosis. Other than proteinuria, the RF-based 271 model revealed that the ESA of serum creatinine level was also ranked in the top-10 in terms of 272 feature importance. Therefore, it was crucial to carefully observe the temporal trends of urinary 273 protein excretion and kidney function to predict GFR decline as early as possible. We consider 274 the aforementioned conclusions of the study to be novel and informative.

i do not understand this!!

Other factors that need to be discussed when evaluating the conclusions is the fact that episodes of AKI can affect the rate of change of kidney function overtime and the fact that stable renal function in a population of this age is very common. Reference Erikson KI 2006 ,Population of similar age where over a 10 year interval 27% had no decline kidney function.Another Issue that needs to be addressed in these types of studies is the competitive risk of death versus decline of kidney function and how that was adjusted for in this analysis

Reply:

Thank you for valuable comments. As you pointed out, AKI is relevant to GFR trajectory, and therefore, we added history of AKI as a variable. According to ICD-10, we examined AKI including acute renal failure, acute tubular injury, acute tubular necrosis, and so on. In both RD and non-RD groups, only around 4.5% of CKD patients had history of AKI. The results of re-analysis after adding history of AKI did not change much (Table 2 and 3). We revised the description in the Discussion (page 20, line 290 – page 21, line 294) and added the references (#28 and #29). Unfortunately, we could not find the article you recommended (we used the keyword of Erikson and KI but could not find)

---

## [Decision Letter · Decision Letter 1]

28 Aug 2020

PONE-D-20-15213R1

Increasing tendency of urine protein is a risk factor for rapid GFR decline in patients with CKD: A machine learning-based prediction model by using a big database.

PLOS ONE

Dear Dr. Inaguma,

Thank you for submitting your manuscript to PLOS ONE. After careful consideration, we feel that it has merit but does not fully meet PLOS ONE’s publication criteria as it currently stands. Therefore, we invite you to submit a revised version of the manuscript that addresses the points raised during the review process.

We look forward to receiving your revised manuscript.

Kind regards,

Tatsuo Shimosawa, M.D., Ph.D.

Academic Editor

PLOS ONE

Reviewers' comments:

Reviewer's Responses to Questions

**Comments to the Author**

1. If the authors have adequately addressed your comments raised in a previous round of review and you feel that this manuscript is now acceptable for publication, you may indicate that here to bypass the “Comments to the Author” section, enter your conflict of interest statement in the “Confidential to Editor” section, and submit your "Accept" recommendation.

Reviewer #1: All comments have been addressed

Reviewer #2: All comments have been addressed

2. Is the manuscript technically sound, and do the data support the conclusions?

Reviewer #1: Yes

Reviewer #2: No

3. Has the statistical analysis been performed appropriately and rigorously? 

Reviewer #1: Yes

Reviewer #2: I Don't Know

4. Have the authors made all data underlying the findings in their manuscript fully available?

Reviewer #1: No

Reviewer #2: Yes

5. Is the manuscript presented in an intelligible fashion and written in standard English?

Reviewer #1: Yes

Reviewer #2: No

6. Review Comments to the Author

Reviewer #1: (No Response)

Reviewer #2: Kidney International (2006) 69, 375–382. doi:10.1038/sj.ki.5000058 ericson reference

it needs further revisions

title change to urine protein is a risk factor for rapid GFR decline in patients

with CKD: A machine learning-based prediction model by using a big database.

remove from line 57with respect to kidney dysfunction during stages 1, 2, and 3a ofCKD.

all mentions of GFR should be eGFR which is what you are using not actual GFR

Meanwhile, use of renin angiotensin system inhibitors was low in the RD group- COMMENT ON THIS IN DISCUSSION

LEAVE OUT Further, glomerular

283 hyperfiltration is known to lead to proteinuria and glomerular sclerosis, and subsequently result

284 in the decline of GFR. Therefore, we attempted to reduce intra-glomerular blood pressure by

285 prescribing medications, including renin angiotensin blockers and sodium glucose transporter-1.

286 The strong correlation between the ESA of proteinuria and GFR decline established in the present

287 study was proved under the assumption that the increase in urinary protein excretion reflected the

288 exacerbation of glomerular hypertension and sclerosis. this is not correct not what you did

7. PLOS authors have the option to publish the peer review history of their article (what does this mean?). If published, this will include your full peer review and any attached files.

Reviewer #1: No

Reviewer #2: No

---

## [Author Response · Author response to Decision Letter 1]

30 Aug 2020

Reply:

Thanks for your comments. 

1. As you pointed out, we removed description from line 57 (with respect to kidney dysfunction during stages 1, 2, and 3a of CKD) and from line 283 to 288 (Further, glomerular hyperfiltration is known to lead to proteinuria and glomerular sclerosis, and subsequently result in the decline of GFR. Therefore, we attempted to reduce intra-glomerular blood pressure by prescribing medications, including renin angiotensin blockers and sodium glucose transporter-1. The strong correlation between the ESA of proteinuria and GFR decline established in the present study was proved under the assumption that the increase in urinary protein excretion reflected the exacerbation of glomerular hypertension and sclerosis.).

2. We changed all “GFR” to “eGFR”.

3. We added the description and the Reference #28.

---

## [Editor Report · Decision Letter 2]

3 Sep 2020

Increasing tendency of urine protein is a risk factor for rapid eGFR decline in patients with CKD: A machine learning-based prediction model by using a big database.

PONE-D-20-15213R2

Dear Dr. Inaguma,

We’re pleased to inform you that your manuscript has been judged scientifically suitable for publication and will be formally accepted for publication once it meets all outstanding technical requirements.

Kind regards,

Tatsuo Shimosawa, M.D., Ph.D.

Academic Editor

PLOS ONE
---

## [Editor Report · Acceptance letter]

8 Sep 2020

PONE-D-20-15213R2 

Increasing tendency of urine protein is a risk factor for rapid eGFR decline in patients with CKD: A machine learning-based prediction model by using a big database. 

Dear Dr. Inaguma:

I'm pleased to inform you that your manuscript has been deemed suitable for publication in PLOS ONE. Congratulations! Your manuscript is now with our production department. 

Kind regards, 

on behalf of

Prof. Tatsuo Shimosawa 

Academic Editor

PLOS ONE